# Reference-based Burst Super-resolution

## ABSTRACT

Burst super-resolution (BurstSR) utilizes signal information from multiple adjacent frames successively taken to restore rich textures. However, due to hand tremors and other image degradation factors, even recent BurstSR methods struggle to reconstruct finely textured images. On the other hand, reference-based super-resolution (RefSR) leverages the high-fidelity reference (Ref) image to recover detailed contents. Nevertheless, if there is no correspondence between the Ref and the low-resolution (LR) images, the degraded output is derived. To overcome the limitations of existing BurstSR and RefSR methods, we newly introduce a reference-based burst super-resolution (RefBSR) that utilizes burst frames and a high-resolution (HR) external Ref image. The RefBSR can restore the HR image by properly fusing the benefits of burst frames and a Ref image. To this end, we propose the first RefBSR framework that consists of Ref-burst feature matching and burst feature-aware Ref texture transfer (BRTT) modules. In addition, our method adaptively integrates features with better quality between Ref and burst features using Ref-burst adaptive feature fusion (RBAF). To train and evaluate our method, we provide a new dataset of Ref-burst pairs collected by commercial smartphones. The proposed method achieves state-of-the-art performance compared to both existing RefSR and BurstSR methods, and we demonstrate its effectiveness through comprehensive experiments. The source codes and the newly constructed dataset will be made publicly available for further research.

## CCS CONCEPTS

• **Computing methodologies → Computational photography**.

## KEYWORDS

Low-level vision, Reference-based super-resolution, Burst super-resolution.

## 1 INTRODUCTION

Burst photography has become popular in recent years with the increasing use of smartphones. Since burst super-resolution (BurstSR) utilizes sub-pixels in multi-frames, high-resolution (HR) images can be obtained. One of the key challenges of BurstSR is to align and fuse sub-pixels between frames into a base frame (*i.e.* the first frame). However, due to different brightnesses or large movements between frames, sub-pixels can be misaligned, leading to incorrect results. On the other hand, reference-based super-resolution (RefSR)

*ACM MM, 2024, Melbourne, Australia*

© 2024 Copyright held by the owner/author(s). Publication rights licensed to ACM.
ACM ISBN 978-x-xxxx-xxxx-x/YY/MM
https://doi.org/10.1145/nnnnnnn.nnnnnnn

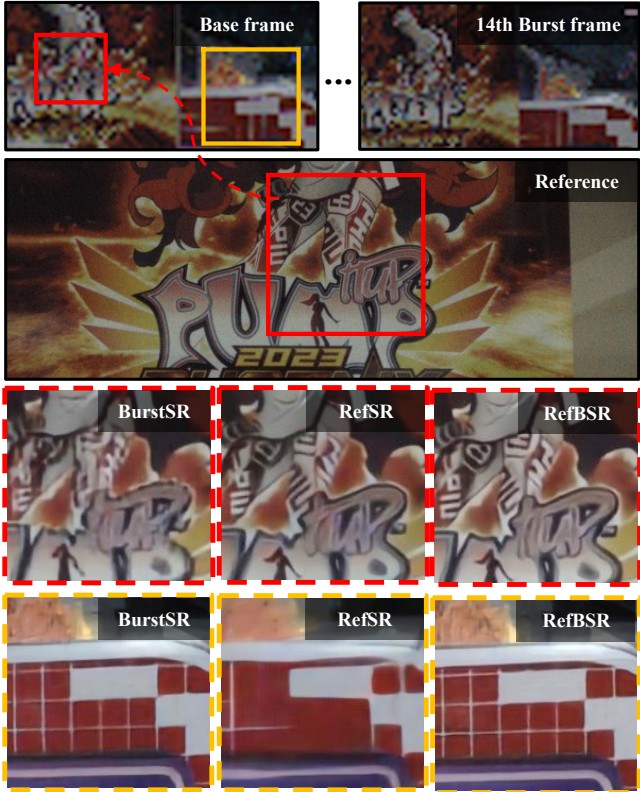

**Figure 1: We visualize the result for each method. (Red dotted box) The matched region between the Ref image and the LR base frame. (Yellow dotted box) The region in the base frame containing no correspondence with the Ref image. When there are matched areas in the Ref image, our RefBSR produces results similar to RefSR, otherwise it produces results similar to BurstSR. For better visualization, we convert the Raw inputs to RGB format.**

aims to reconstruct the low-resolution (LR) image with an extra HR reference (Ref) image. The Ref image has a similar scene to the LR image, but it is captured with different environments such as equipment, location, and time. One of the core ideas in the RefSR task is to align the Ref image based on the similarities between the Ref and LR images. However, the results are degraded due to inaccurate matching between the Ref and LR images. As shown in Figure 1, the red dotted box in the Ref image corresponds to the red dotted box in the LR frame. In the yellow dotted box in the LR frame, there are no correspondences with the Ref image. As shown in the RefSR result of the red dotted box, in the case there are overlapping regions, the RefSR generates high-frequency textures by transferring the detailed textures in the Ref image to the LR frame. In contrast, the results of BurstSR suffer from a lack of detail. On the other hand, in case of failed matching, the performance of

the RefSR method is degraded as shown in the yellow dotted box. If the RefSR method is less helpful due to failed matching, sub-pixels of the burst frame can be utilized to recover the region as shown in the yellow dotted box of the BurstSR result. These examples demonstrate the advantages and disadvantages of each technique.

Based on the above observations, the goal of this paper is to combine the strengths of each method and mitigate the weaknesses of them. In this regard, we introduce a new unexplored task: reference-based burst super-resolution (RefBSR) that integrates the two methods. The outcomes of RefBSR showcased in the red dotted box in Fig. 1 preserve details of the window textures present in the building. These results closely match the detailed outcomes obtained from RefSR. On the other hand, the results of RefBSR presented in the yellow dotted box show the textural details in the exterior of the building, similar to the detailed results of BurstSR. Therefore, our proposed RefBSR integrates the high-frequency texture from the Ref image and the multiple clues from the sub-pixels in the burst frames.

To achieve a suitable fusion of the two approaches, we propose the burst feature-aware Ref texture transfer (BRTT) module. The BRTT module aligns the Ref feature using each burst feature individually into the grid of the base frame. At this point, Ref-burst adaptive feature fusion (RBAF) is utilized to fuse the Ref and burst features with adaptive weighting. Then, each Ref-burst feature is fused together by the decoder to produce the final result. We have the advantage of sending the Ref feature to every burst feature to generate different alignment results. To train and test the proposed RefBSR task, we provide a new RefBSR dataset. Using a commercial smartphone camera, we collect a total of 2,287 pairs of datasets containing burst frames and Ref images. The burst frames consist of a natural hand tremor and the Ref image contains similar scenes to the burst frames. Specifically, the Ref images are captured at a variety of times and locations. We separate 2,002 pairs of images into a training set and 285 pairs of images into a test set. Our contributions are summarized as follows.

- We introduce the first framework for reference-based burst super-resolution (RefBSR).
- For the new RefBSR task, we design the burst feature-aware Ref texture transfer (BRTT) and Ref-burst adaptive feature fusion (RBAF) modules to adaptively fuse the Ref and the burst features.
- We provide the RefBSR dataset containing pairs of RAW burst frames and Ref images for training and testing.

## 2 RELATED WORKS

### 2.1 Reference-based Super-Resolution

Reference-based super-resolution (RefSR) [9, 10, 18, 24, 27, 31–34, 36–40], focuses on restoring super-resolution images by transferring the textures of additional Ref images. Useful textures in the Ref image can be extracted by matching with the LR image, and it is important to precisely align the matched Ref features. Zheng *et al.* [39] suggested CrossNet estimates the optical flow between the Ref and LR images and exploits multi-scale warping. To improve matching performance, Zhang *et al.* [38] proposed SRNTT that adopts a patch-matching approach. Furthermore, for better feature matching and alignment, Yang *et al.* [32] proposed TTSR based on

a learnable texture transformer while Shim *et al.* [24] introduced SSEN using a deformable convolution. For improving matching efficiency, Lu *et al.* [18] proposed MASA consisting of coarse-to-fine matching mechanisms. Based on the knowledge distillation and contrastive learning, Jiang *et al.* [10] suggested $C^2$-Matching robust to the rotation and scale variations in the matching. Cao *et al.* proposed DATSR [4] that replaces the previous multi-scale structure with the U-net structure and applied the Swin transformer [16]. Also, Huang *et al.* proposed that TDF [9] separates the two networks into the SR network and RefSR network to mitigate misalignment and misuse issues. Recently, Zhang *et al.* proposed that MRefSR [34] utilizes the feature mechanism to selectively transfer the alignment of the Ref image. Although RefSR is able to use the textures of the Ref image to generate a more detailed result, its performance in the regions that are not similar to the Ref image is inevitably degraded.

### 2.2 Burst Super-Resolution

In contrast to RefSR, burst super-resolution (BurstSR) [1–3, 6–8, 12, 14, 19–22] exploits multiple shifted LR frames that contain the same contents to recover the high-quality image. Therefore, it is important to align features from multi-frames and aggregate them in the BurstSR task. In recent years, many BurstSR approaches based on deep learning have been proposed. For instance, Deudon *et al.* proposed HighResNet [6] using multi-frames for remote sensing images. Also, Bhat *et al.* [1] proposed DBSR based on PWC [26] network for the feature alignment and aggregation using attention weights. For improving the performance, MFIR [2] using the deep reparameterization of MAP and EBSR [20] based on the PCD with multi-scales mechanism were suggested. Furthermore, Luo *et al.* [19] introduced BSRT based on an optical flow estimator to align more robustly with PCD [28] module. Recently, Mehta *et al.* [22] proposed GMTNet that utilizes the attention mechanism to generate the offset for better alignment with the deformable convolution. Dudhane *et al.* [8] suggested Burstormer which utilizes local and non-local information at multi-scale to improve alignment. However, BurstSR can lead to incorrect results due to the large movement in the scene. Furthermore, the resolution and quality of RAW burst frames are limited. These drawbacks can be overcome by utilizing techniques in RefSR, which can exploit high-frequency textures from non-local and global regions. Therefore, in this paper, we introduce a new reference-based burst super-resolution (RefBSR) task to effectively fuse the Ref image and burst frames to restore a high-quality image.

## 3 METHODS

### 3.1 Reference-Based Burst SR Framework

Reference-based burst super-resolution (RefBSR) aims to recover a high-resolution image using burst frames and an external reference (Ref) image. To achieve this, we newly introduce a RefBSR framework that integrates the burst frames and the Ref image. As illustrated in Figure 2, our model takes a RAW Ref image $I_r$ and burst frames $I_b = \{I_{b_i}\}_{i=1}^{N}$ as inputs, where $N$ is the number of burst frames. Then, it produces a super-resolved output $I_{sr}$ that has the same viewpoint as the first image (*i.e.* base frame) $I_{b_1}$ in the burst frame set $I_b$.

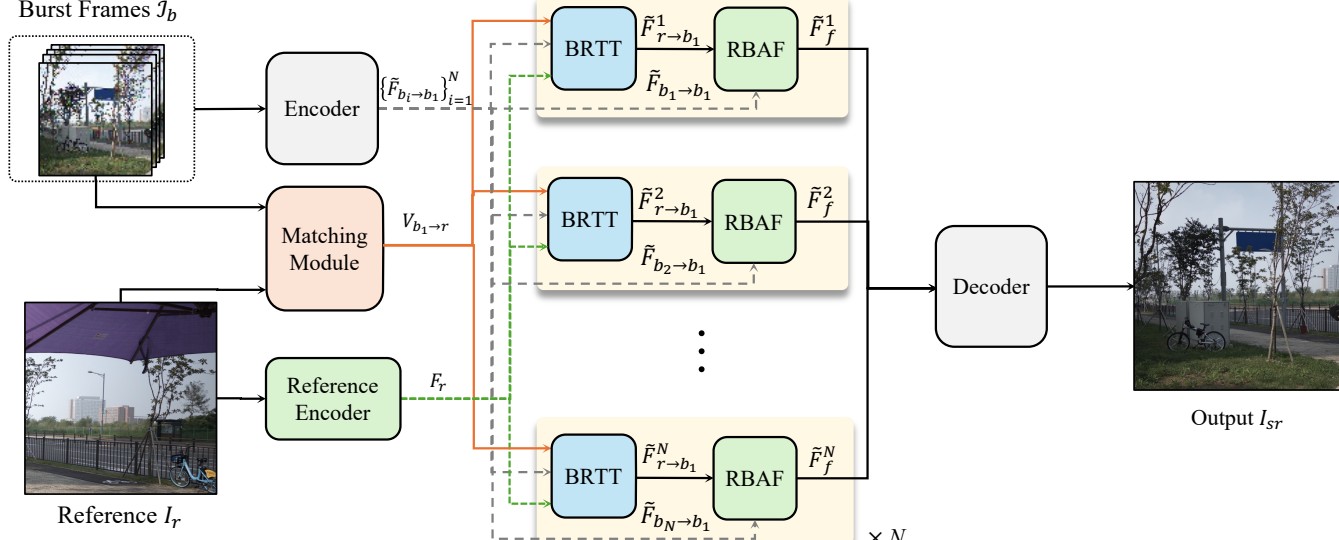

**Figure 2: Overall architecture of reference-based burst super-resolution framework. Our proposed model takes burst frames with the reference image and generates the super-resolved image. For better visualization, inputs and output are converted to RGB images.**

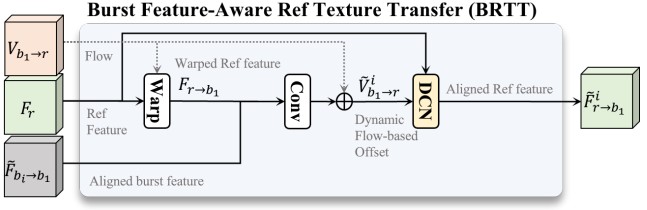

**Figure 3: Illustration of the burst feature-aware Ref texture transfer (BRTT) module. The BRTT module takes the flow field $V_{b_1 \rightarrow r}$, Ref feature $F_r$, and one aligned burst feature $\tilde{F}_{b_i \rightarrow b_1}$ to yield the burst feature-aware aligned Ref feature $\tilde{F}^i_{r \rightarrow b_1}$.**

In Section 3.2, we first perform the base and reference frame matching to obtain a flow field $V_{b_1 \rightarrow r}$ from the base frame to the reference frame. In Section 3.3, we obtain burst feature-aware aligned Ref features $\{\tilde{F}^i_{r \rightarrow b_1}\}^N_{i=1}$ by transferring a Ref feature to the base frame based on burst frame features using the BRTT module. The Ref feature $F_r$ is extracted from $I_r$ by VGG16 [25], while $N$ burst features $\{\tilde{F}_{b_i \rightarrow b_1}\}^N_{i=1}$, which are aligned to the base frame, are extracted from $\mathcal{I}_b$ by FG-DCN [5, 19]. In Section 3.4, we develop the Ref-burst adaptive feature fusion to combine $\{\tilde{F}^i_{r \rightarrow b_1}\}^N_{i=1}$ and $\{\tilde{F}_{b_i \rightarrow b_1}\}^N_{i=1}$ effectively. Last, the decoder generates the super-resolved frame $I_{sr}$ from the fused feature.

## 3.2 Base and Reference Frame Matching

In order to transfer the texture from the Ref image $I_r$ to the low-resolution base frame $I_{b_1}$ effectively, it needs to find correspondences between $I_r$ and $I_{b_1}$. We first obtain a base frame matching feature $M_{b_1} = \phi(I_{b_1})$ and a Ref matching feature $M_r = \phi(I_r)$ by employing a feature extractor $\phi(\cdot)$ in [23]. Then, we compute the patch-based correlation between features $M_{b_1}$ and $M_r$ to obtain the flow filed $V_{b_1 \rightarrow r}$ for matching from the base frame to the Ref frame. The correlation at pixel $\mathbf{x}$ for a flow vector $\mathbf{v}$ is defined as

$$C(\mathbf{x}, \mathbf{v}) = \sum_{\mathbf{p} \in [-k,k] \times [-k,k]} M_{b_1}(\mathbf{x} + \mathbf{p})^T M_r(\mathbf{x} + \mathbf{p} + \mathbf{v}), \quad (1)$$

where $2k + 1$ is a patch size. Thus, $C(\mathbf{x}, \mathbf{v})$ denotes the correlation between patches centered at $\mathbf{x}$ in $M_{b_1}$ and centered at $\mathbf{x} + \mathbf{v}$ in $M_r$. Note that we compute all correlations between every pixel in the base and Ref frames for global matching. Then, the flow vector $V_{b_1 \rightarrow r}(\mathbf{x})$ for pixel $\mathbf{x}$ is defined as

$$V_{b_1 \rightarrow r}(\mathbf{x}) = \arg\max_{\mathbf{v}}(C(\mathbf{x}, \mathbf{v})). \quad (2)$$

## 3.3 Burst Feature-Aware Ref Texture Transfer

The next step is to transfer the rich textures in the Ref image $I_r$ into the grid of the base frame $I_{b_1}$. For this purpose, we introduce the burst feature-aware Ref texture transfer (BRTT) that exploits the sub-pixel information in burst features for texture transfer of the Ref frame. The BRTT module takes the flow filed $V_{b_1 \rightarrow r}$, Ref feature $F_r$, and one of the aligned burst features $\tilde{F}_{b_i \rightarrow b_1}$ as inputs, as shown in Figure 3.

BRTT first obtains a flow-based aligned Ref feature for each pixel $\mathbf{x}$ through backward warping, which is expressed as

$$F_{r \rightarrow b_1}(\mathbf{x}) = F_r(\mathbf{x} + V_{b_1 \rightarrow r}(\mathbf{x})). \quad (3)$$

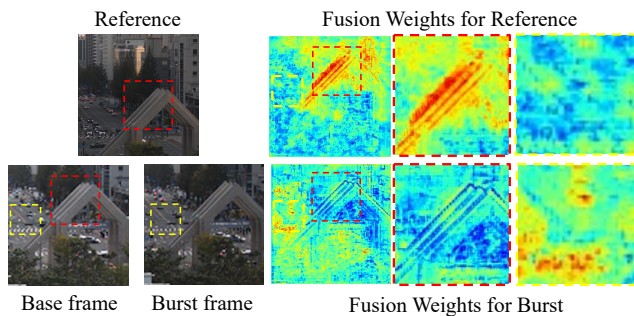

Reference    Fusion Weights for Reference

Base frame    Burst frame    Fusion Weights for Burst

**Figure 4: Visualization of the Ref and burst weights. The correspondence between the Ref and base frame is illustrated as the red box. No overlapping region is represented with the yellow box.**

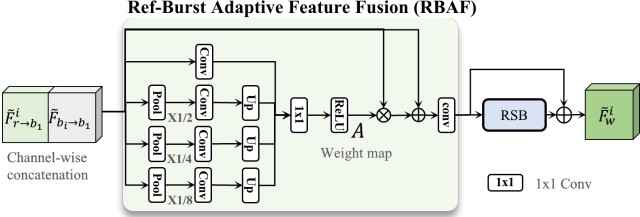

**Figure 5: Illustration of the Ref-burst adaptive feature fusion module. The adaptive feature weight map $A$ is generated to fuse aligned Ref and burst features effectively.**

Here, $F_r(\mathbf{x} + V_{b_1 \to r}(\mathbf{x}))$ is computed by bilinear interpolation. Considering that some pixels in the base frame may not be well matched to the Ref frame, we refine the flow $V_{b_1 \to r}^i$ by exploiting the aligned burst feature $\tilde{F}_{b_i \to b_1}$. Thus, $F_{r \to b_1}$ and $\tilde{F}_{b_i \to b_1}$ are concatenated and processed by a $3 \times 3$ convolution layer to obtain the dynamic flow-based offset (DFO) $\tilde{V}_{b_1 \to r}^i$, which is given by

$$\tilde{V}_{b_1 \to r}^i = \text{Conv}([F_{r \to b_1}, F_{b_i \to b_1}]) + V_{b_1 \to r}, \qquad (4)$$

where $\text{Conv}(\cdot)$ and $[\cdot, \cdot]$ denote the convolution and concatenation operations. We adopt a deformable convolution to generate a burst feature-aware aligned Ref feature

$$\tilde{F}_{r \to b_1}^i = \text{DCN}(F_r, \tilde{V}_{b_1 \to r}^i) \qquad (5)$$

where DCN is a function for the deformable convolution and the dynamic flow-based offset $\tilde{V}_{b_1 \to r}^i$ is used in the deformable convolution. BRTT is repeated for each burst feature, and thus burst feature-aware aligned Ref features $\{\tilde{F}_{r \to b_1}^i\}_{i=1}^N$ are obtained.

### 3.4 Ref-Burst Adaptive Feature Fusion

We combine the aligned Ref and burst features $\{(\tilde{F}_{r \to b_1}^i, \tilde{F}_{b_i \to b_1})\}_{i=1}^N$ to integrate the information of high-frequency texture in the Ref image and multiple clues of the sub-pixels in the burst frames. Even though the two features can be simply concatenated for the feature fusion, we expect the feature fusion to be enhanced by assigning

adaptive weights along feature channels for each position. For instance, in Figure 4, a crosswalk in the yellow box in the base frame is not well visible in the Ref frame unlike another burst frame. For those regions, it is reliable to assign higher weights to the burst feature than the Ref feature. In contrast, visible regions such as the red box should contain higher weights on the Ref feature to exploit the high-frequency texture information in the Ref frame.

For the adaptive feature fusion of Ref and burst features, we design multi-scale convolution layers to estimate adaptive fusion weights as illustrated in Figure 5. Specifically, Ref and burst features in each pair are concatenated, i.e. $[\tilde{F}_{r \to b_1}^i, \tilde{F}_{b_i \to b_1}]$, and the concatenated feature is fed into four branches, where the four branches deal with different scales (1, 1/2, 1/4, and 1/8 scales). Then, we concatenate four branch outputs and apply the point-wise convolution followed by an activation function to generate the adaptive fusion weight map $A$. Fig. 4 visualize the adaptive fusion weight map for the Ref frame and one of the burst frame. Here, adaptive fusion weights for the Ref feature are computed by averaging weights along the first half of the channels in $A$, while the other half of the channels are used for the burst frame. As illustrated in Fig. 4, positions near a sculpture in the red box, where it is visible in the Ref frame, contain higher weights in the Ref frame than the burst frame. On the other hand, invisible regions such as a crosswalk in the pink box provide lower weights for the Ref frame.

Next, we compute a weighted feature $\tilde{F}_w^i$ using the adaptive fusion weight map $A$ as

$$\tilde{F}_w^i = A \otimes [\tilde{F}_{r \to b_1}^i, \tilde{F}_{b_i \to b_1}] + [\tilde{F}_{r \to b_1}^i, \tilde{F}_{b_i \to b_1}] \qquad (6)$$

where $\otimes$ denotes an element-wise multiplication operation. To this end, the weighted feature $\tilde{F}_w^i$ is processed by a convolution layer and a residual block to yield a fused Ref-burst feature $\tilde{F}_f^i$. We repeat the adaptive feature fusion process for all pairs $\{(\tilde{F}_{r \to b_1}^i, \tilde{F}_{b_i \to b_1})\}_{i=1}^N$, resulting in $N$ fused features $\{\tilde{F}_f^i\}_{i=1}^N$. Finally, the $N$ fused features are concatenated and processed by the decoder to produce the super-resolved frame $I_{sr}$. A detailed explanation of the decoder is in the supplementary material.

### 3.5 RefBSR Dataset

To accomplish our goal of reference-based burst super-resolution, we construct a new dataset, called RefBSR, for the training and testing stages.

For dataset construction, we first capture 232 pairs of RAW Ref and burst images using the custom camera app in iPhone 14 pro. Specifically, we obtain 14 burst frames for each sample using continuous shooting. To collect Ref images, we take a picture of scenes with different locations, viewpoints, and time from the burst frames. Unlike the BurstSR dataset [1], we collect the dataset by hand without a tripod for realistic hand tremors. In other words, some burst frames in our dataset may contain motions with each other due to hand tremors.

Since the captured images are in RAW format, we transform the first burst frames and Ref images into 3-channel color images through demosaicking, where the transformed first burst frames are used as the ground truth (GT). For LR images, the burst frames are downsampled $\times 1/4$ by bicubic interpolation with the Bayer

Table 1: Qualitative comparison of the proposed method with SISR, RefSR, BurstSR, and RefBSR methods. † indicates the feature extractor [10] is replaced with [23]. The best result is boldfaced.

|  | | Synthetic dataset | | | RefBSR dataset | | |
|---|---|---|---|---|---|---|---|
|  | Method | PNSR | SSIM | LPIPS | PSNR | SSIM | LPIPS |
| SISR | Bicubic | 27.24 | 0.709 | 0.311 | 36.19 | 0.878 | 0.133 |
|  | SwinIR [15] | 37.39 | 0.922 | 0.108 | 42.97 | 0.963 | 0.082 |
| RefSR | $C^2$-Matching [10] | 39.15 | 0.946 | 0.066 | 42.23 | 0.956 | 0.076 |
|  | MASA [18] | 41.63 | 0.965 | 0.043 | 43.56 | 0.965 | 0.066 |
|  | DATSR [4] | 43.36 | 0.977 | 0.023 | - | - | - |
|  | MRefSR [34] | 42.87 | 0.973 | 0.023 | 42.74 | 0.960 | 0.068 |
|  | MRefSR$^\dagger$ | 43.02 | 0.976 | 0.022 | 43.63 | 0.965 | 0.057 |
| BurstSR | DBSR [1] | 39.17 | 0.946 | 0.081 | 44.23 | 0.969 | 0.076 |
|  | MFIR [2] | 41.56 | 0.964 | 0.045 | 45.04 | 0.973 | 0.054 |
|  | RBSR [30] | 42.44 | 0.970 | 0.034 | 45.88 | 0.977 | 0.047 |
|  | Burstormer [8] | 42.83 | 0.970 | - | 43.83 | 0.968 | 0.031 |
| RefVSR | RefVSR [13] | 39.88 | 0.954 | 0.062 | 44.25 | 0.969 | 0.067 |
| RefBSR | Ours | **44.21** | **0.978** | **0.020** | **46.49** | **0.980** | **0.030** |

pattern, resulting in the scale factor ×8. To construct pairs of Ref and burst images for the new dataset, the Ref and GT images are divided into 640×640 patches, while the downsampled burst frames are divided into 80×80 patches. Then, we only take pairs that satisfy the sufficient similarity between the GT and Ref images. For this purpose, we adopt the SIFT [17] algorithm to extract the features from the GT and Ref images and perform the brute force matching between a patch from GT and all patches from the Ref image. We then discard patch pairs whose matched features are fewer than 50. To this end, our new dataset consists of 2,287 pairs of Ref and burst images, which are divided into 2,002 training and 285 testing pairs.

## 4 EXPERIMENTS

### 4.1 Training Details

**Datasets and Evaluation.** For both training and testing, we use the representative synthetic burst dataset [1], as well as our RefBSR dataset as introduced in Section 3.5. Given that the existing synthetic burst dataset [1] exclusively provides burst frames, we generate the synthetic Ref images. We apply a random perturbation to the GT images between 0-20 to create Ref images as introduced in [10]. The synthetic dataset provides input burst frames of size 4×96×96 with a Bayer pattern and Ref images of size 3×384×384. The size of GT image is also 3×384×384. In consequence, we utilize 46,839 pairs for the training dataset and 300 pairs for the testing set. We employ PSNR, SSIM [29], and LPIPS [35] metrics for evaluation. Note that all methods are evaluated on the linear sensor space. Results are evaluated on the linear sensor for the metrics.

**Implementation details.** To ensure smooth training, we first train the proposed network with the synthetic dataset, and then train it with the RefBSR dataset. We set epoch to 80 and batch size to 16 for training with the synthetic dataset. We select ADAM [11] as the optimizer and set the learning rate to 1e-4 and the betas to $\beta_1 = 0.9$ and $\beta_2 = 0.99$. For training with the RefBSR dataset, we opt for the same setting with the optimizer, learning rate and betas as the synthetic dataset, while we set epoch to 130 and batch size

to 8. The proposed model is trained using the $L_1$ loss

$$L_1 = ||I_{sr} - I_{gt}||_1 \tag{7}$$

where $I_{sr}$ and $I_{gt}$ denote the super-resolved result and the ground-truth, respectively. Our method is trained for about 3 to 4 days with A6000 GPUs.

### 4.2 Comparisons with Existing Methods

**Quantitative Comparison.** We compare our method with the SISR [15], RefSR [10, 34], BurstSR [1, 2, 8], and RefVSR [13] methods. Note that RefBSR is a newly unexplored task, and thus we pick the state-of-the-art SISR, RefSR, and BurstSR, RefVSR to validate the effectiveness of the proposed network. For fairness, all existing methods are firstly trained with the synthetic dataset and then trained on the RefBSR dataset as done in our model training. We train SISR methods using the base frame only, Also, RefSR methods are trained with the base and Ref images, while BurstSR methods are trained with all burst frames.

Table 1 shows quantitative results on the synthetic and the Ref-BSR datasets. In the RefSR method, $C^2$-Matching [10] and LMR [34] provide a lower performance than SwinIR [15] on the RefBSR dataset, and we assume that the feature extractor for matching in [10] degrades the performance. LMR$^\dagger$ indicates another experiment with different feature extractor [23] for matching. When the feature extractor in LMR is replaced with the more effective feature extractor [23], LMR$^\dagger$ outperforms SwinIR on the RefBSR dataset. In BurstSR methods, since Burstormer [8] does not utilize the flow estimator, it yields a lower performance than the previous BurstSR models [1, 2] on the RefBSR dataset, which contains realistic movements. Finally, our proposed method outperforms the existing methods for all metrics on both synthetic and RefBSR datasets. This indicates that our model effectively combines the high-frequency texture from the Ref image and the multiple sub-pixel cues in the burst frames. Comparing our RefBSR model to the second best method MFIR [2], we achieve a performance improvement of over 1.45dB in terms of PSNR on the RefBSR dataset.

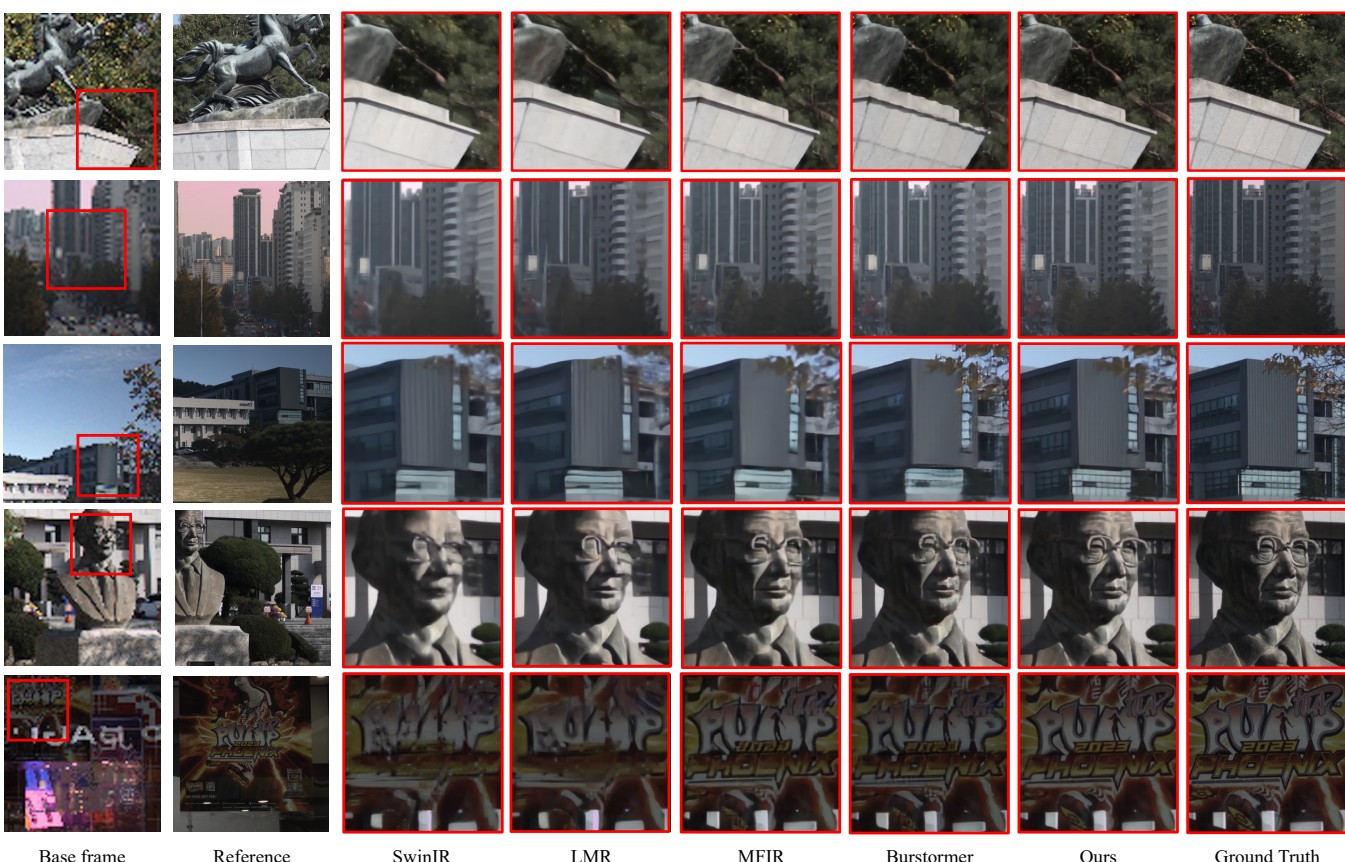

| Base frame | Reference | SwinIR | LMR | MFIR | Burstormer | Ours | Ground Truth |

**Figure 6: Qualitative comparison with SwinIR [15], LMR [34], MFIR [2], Burstormer [8], and on RefBSR dataset. The results of our model have more detailed content. The base frame and the Ref image are converted to RGB for better visualization.**

**Qualitative Comparison.** Figure 6 shows qualitative results of the proposed method, and the state-of-the-art models including SwinIR [15], LMR [34], and MFIR [2] on the RefBSR dataset. Our model faithfully recovers detailed textures by fusing multiple LR frames and the Ref image. For instance, in the 3rd row, even though the Ref image and LR base frame contain different brightness, the textures of the structure in our model are visible more clearly than in other models. Also, in the last row, our result contains high-fidelity letters, *e.g.*'2023', as compared with other methods.

## 4.3 Ablation Study

To demonstrate the effectiveness of our proposed model, we perform ablation studies on the synthetic and the RefBSR dataset. In Table 2, we design the baseline model *Base* without the aligned burst feature in (4) in the BRTT module and without the RBAF module. Thus, in *Base*, only Ref feature is used to update the flow filed $V_{b_1 \to r}$ in (4) without the information of burst features. Also, in *Base*, we concatenate the aligned burst and Ref features and pass it to the RSB blocks without the RBAF module. Next, *DFO+Base* indicates that the dynamic flow-based offsets (DFO) are included in *Base*, which is equivalent to the proposed model without the RBAF module. Finally, *RBAF+DFO+Base* is the final version of our model.

**Table 2: Ablation study for the proposed BRTT and RBAF modules. *Base*, *DFO* and *RBAF* indicate the baseline, the dynamic flow-based offset with the aligned burst feature in BFTT, and the RBAF module, respectively. The best result is boldfaced.**

| Dataset | Synthetic Dataset | | | RefBSR Dataset | | |
|---|---|---|---|---|---|---|
| Method | PNSR | SSIM | LPIPS | PNSR | SSIM | LPIPS |
| *Base* | 43.72 | 0.976 | 0.022 | 46.33 | 0.977 | 0.033 |
| *DFO + Base* | 43.94 | 0.977 | 0.021 | 46.41 | 0.978 | 0.031 |
| *RBAF + DFO + Base* | **44.21** | **0.978** | **0.020** | **46.49** | **0.980** | **0.030** |

In Table 2, we observe that the performance gradually increases on the synthetic and RefBSR datasets as we add each module one by one. Further analysis of each method is detailed in the following explanations.

**Analysis on BRTT Module.** We assert that various aligned reference features can be obtained by exploiting the burst features, and those burst feature-aware Ref features transfer rich textures in the Ref image to the base frame effectively. In Tab. 2, when we add the aligned burst features to generate the dynamic flow-based offsets (*DFO+Base*), there is a performance improvement against

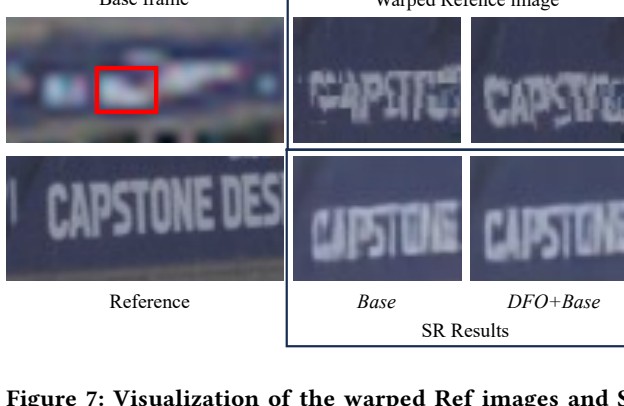

Figure 7: Visualization of the warped Ref images and SR results using *Base* and *DFO+Base.*

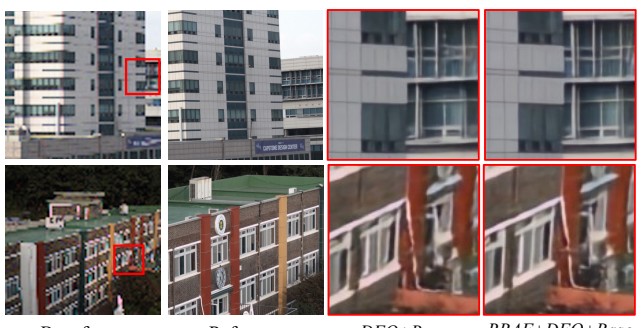

Figure 8: SR results of the proposed method with and without the RBAF module. By using the RBAF module, detailed and structural textures are better restored.

*Base* on both synthetic and RefBSR datasets. Also, Figure 7 compares warped Ref images (top row) and SR results (bottom row) of *Base* and *DFO+Base.* The warped Ref images are obtained by backward warping from the Ref image to the base frame using estimated offsets in (4). We can observe that the textures of the letter are well restored by the *DFO+Base* than *Base* on both warping and SR results. It indicates that the proposed BRTT yields more accurate offsets by exploring burst features, resulting in visually pleasing SR results.

**Analysis on RBAF Module.** By applying the adaptive weights to the aligned Ref feature and the aligned burst feature, the RBAF module ensures that the two features are fused properly. In Tab. 2, we verify the effectiveness of the RBAF module in terms of quantitative results on both datasets. As shown in Figure 8, when we add the RBAF module to *DFO+Base*, *RBAF+DFO+Base* exhibits more faithful restoration. Specifically, the structure of the windows of *Base+DFO+RBAF* on two scenes is clearly reconstructed. These visual results, as well as the quantitative results, demonstrate that the proposed RBAF module focuses more on the important information in each feature.

Table 3: Ablation study of fusion method. The individual fusion method outperforms the group fusion method. The best result is boldfaced.

| Method | Synthetic dataset | | | RefBSR dataset | | |
|---|---|---|---|---|---|---|
| | PSNR | SSIM | LPIPS | PSNR | SSIM | LPIPS |
| Group fusion | 43.81 | 0.976 | 0.022 | 46.03 | 0.976 | 0.035 |
| Individual fusion | **44.21** | **0.976** | **0.020** | **46.49** | **0.980** | **0.030** |

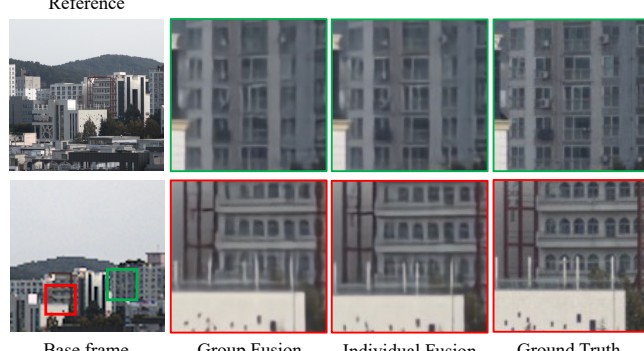

Figure 9: Qualitative comparisons between the group fusion and the individual fusion methods. The individual fusion method provides more detailed textures.

## 4.4 Further Analysis

**Approach of Ref-Burst Fusion.** To combine the aligned burst and the Ref features, it is an issue "how" to integrate burst and Ref features. One of the naive approaches is that all burst features are concatenated and then the Ref feature is fused to the simply aggregated burst feature. We refer to this approach as the group-fusion method. Specifically, in the group-fusion method, the aligned burst feature in (4) for BRTT and (6) for RBAF is replaced with the simply aggregated burst feature. Another option is the individual-fusion method that fuses the Ref feature to each burst feature, which is adopted in our model. Table 3 quantitatively compares the SR performance of the group and individual fusion methods. We see that the individual fusion approach is more effective than the group method. The advantage of individual fusion over group fusion is that we can obtain dynamic flow-based offsets using individual burst features. In contrast, the group fusion provides only one offset for the alignment of Ref image. These dynamic flow-based offsets work well to exploit sub-pixel information in multiple burst frames, and thus remarkable performance is achieved. Furthermore, Figure 9 illustrates that the individual fusion method produces relatively sharper results than the group method. These observations demonstrate that the individual fusion method is more effective than the group fusion method.

**Effect of Reference and Burst Frame.** To validate our RefBSR framework, we investigate experiments without the Ref image or the burst frames. First, we design the modified version of our model without the Ref image. Therefore, the modified model serves as the BurstSR model. Second, we construct our model without the burst

**Table 4: Further analysis on RefSR, BurstSR and RefBSR for our model. BurstSR mode is the model that only takes burst frames as input. RefSR mode is the model that takes the base frame and the Ref image. RefBSR mode is the proposed framework that uses both burst frames and Ref image. The best result is boldfaced.**

|  | Synthetic dataset | | | RefBSR dataset | | |
|---|---|---|---|---|---|---|
| Mode | PSNR | SSIM | LPIPS | PSNR | SSIM | LPIPS |
| BurstSR | 42.23 | 0.968 | 0.035 | 45.46 | 0.976 | 0.048 |
| RefSR | 42.04 | 0.968 | 0.034 | 43.64 | 0.965 | 0.066 |
| RefBSR | **44.21** | **0.978** | **0.020** | **46.49** | **0.980** | **0.030** |

**Table 5: Quantitative results based on the similarity of between the base and Ref images. We divide samples into three similarity levels according to the random perturbation. The best result is boldfaced.**

|  | Synthetic dataset | | |
|---|---|---|---|
| Similarity Level | PSNR | SSIM | LPIPS |
| Hard (20-30) | 43.64 | 0.975 | 0.024 |
| Medium (10-20) | 44.03 | 0.977 | 0.021 |
| Easy(0-10) | **44.37** | **0.978** | **0.020** |

frames. In this case, the model without burst frames takes only a single base frame and the Ref image, which is equivalent to RefSR. In Table 4, we observe that the proposed RefBSR framework, which fuses the RefSR and BurstSR, achieves the best performance. This indicates that our RefBSR framework effectively combines the two approaches.

**Effect of Similarity between Base and Reference Images.** We analyze the effect of the similarity between the base and Ref images. When we construct the synthetic dataset as described in Section 4.1, we employ a random permutation to the ground-truth image for generating Ref images. In the Ref image generation process, we divide the samples into three similarity levels according to the permutation by following [10]: easy (0-10), medium (10-20), and hard (20-30). Table 5 shows that the performance of our model gradually increases as the similarity between base and Ref images increases. This trend demonstrates that our model exploits the reference information effectively.

**Dual Camera Setting.** Our model readily extends to the Dual lens setting. We conduct analyses on dual rigs. For this system, we collect a total of 72 sets of datasets by configuring bursts to wide-angle and utilizing Ref as telephoto image. In the dual camera setting, RefSR encounters a challenge due to the absence of correspondences between the wide-angle as the burst frames and the telephoto as the Ref image, leading to a degradation in quality. One plausible approach to address this issue is to consider employing burst frames in regions lacking correspondences. As demonstrated in Table 6, our model exhibits superior performance compared to previous RefSR, BurstSR, RefVSR method. Furthermore, in the Figure 10, we observe remarkably sharp results in the areas overlapping with Ref. Moreover, in the non-overlapping regions with Ref, there is sufficient utilization of burst frames to capture detailed information.

**Table 6: Quantitative results based on dual camera setting. The best result is boldfaced.**

|  | Dual camera dataset | | |
|---|---|---|---|
| Method | PSNR | SSIM | LPIPS |
| MASA [18] | 45.89 | 0.983 | 0.027 |
| LMR [34] | 45.65 | 0.981 | 0.029 |
| DBSR [1] | 47.58 | 0.988 | 0.027 |
| MFIR [2] | 48.51 | 0.990 | 0.017 |
| Burstormer [8] | 47.64 | 0.989 | 0.020 |
| RefVSR [13] | 47.45 | 0.987 | 0.021 |
| Ours | **50.75** | **0.993** | **0.009** |

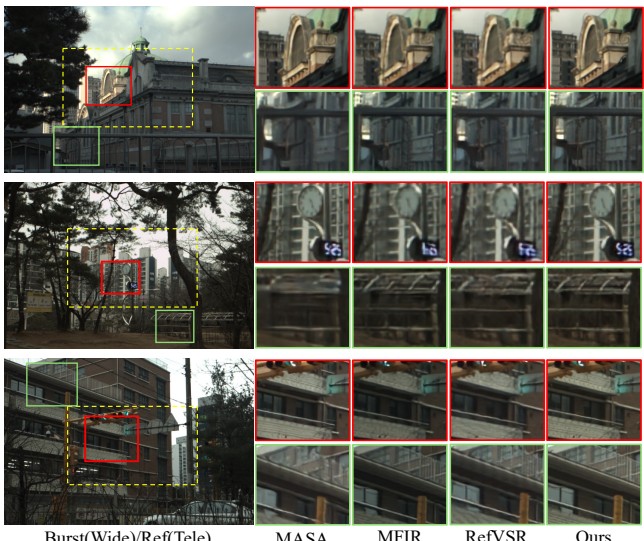

Burst(Wide)/Ref(Tele)    MASA    MFIR    RefVSR    Ours

**Figure 10: Qualitative comparisons on dual camera setting. The yellow dotted box indicates the tele-photo (Ref). The entire image represents the wide-angle photo (burst).**

These results demonstrate that our model operates robustly in dual camera system.

## 5 CONCLUSION

We introduce a new framework for the reference-based super-resolution (RefBSR) with a new dataset. The proposed RefBSR framework reconstructs the super-resolved image using the low-resolution burst frames and the reference image as the inputs. For this purpose, the burst feature aware Ref texture transfer (BRTT) module conveys the fine texture in the reference image to each burst feature. To achieve the reference alignment, we utilize the burst feature to obtain the wide range of the aligned reference feature in the BRTT module. The ref-burst adaptive feature Fusion (RBAF) gives the attention weights to each aligned burst-reference feature and fuses them. We provide a new dataset for the RefBSR task consisting of pairs of the RAW burst frames, the reference image and the ground truth image. Extensive experiments verify our framework and modules on the proposed dataset.

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
