# OpenReview forum: "Reference-based Burst Super-resolution"
_acmmm.org/ACMMM/2024/Conference — MM2024 Poster_

### Official Review · Reviewer_pL2r · 2024-05-23

**Rating:** 4
**Confidence:** 3

**Summary:**

The paper proposes a novel reference-based burst super-resolution method (RefBSR) that combines the advantages of burst frames and high-resolution reference images to significantly enhance image reconstruction quality, and demonstrates its superior performance through comprehensive experiments. The paper also introduces a dataset based on images captured by smartphones.

**Strengths:**

+The proposed dataset is highly practical, and the method is both practical and effective.

+The experimental results in the paper are robust, and the experiments are thorough.

**Limitations:**

-Why not evaluate on the RealMCVSR dataset?

-Why does RefVSR perform worse than non-ref burst SR methods?

-In reference-based SR tasks, aligning the reference frame with other frames is crucial. The paper should provide more analysis on whether the improvements come from the alignment process itself or from the implicit alignment achieved by the feature fusion module.

-Although I find the method in the paper to be quite engineering-focused, the challenge in reference-based approaches lies in the selection and alignment of the references. The paper does not provide sufficient insight into this point. However, the proposed dual-camera dataset and experimental results may advance the development in this direction.

**Suitability:**

3

---

### Official Review · Reviewer_KB84 · 2024-05-24

**Rating:** 5
**Confidence:** 4

**Summary:**

The authors combine reference-based and burst super-resolution to utilize burst frames and a high-resolution (HR) external Ref image, which achieves state-of-the-art performance compared to both existing RefSR and BurstSR methods. Further, they provide a new dataset of Ref-burst pairs collected by commercial smartphones for training and testing.

**Strengths:**

Reasonable structure and novel idea. Solid dataset and evaluation. Sufficient reproducibility and practicality.

**Limitations:**

1. In 3.2, the authors estimate flow between low-resolution base frame I_{b1} and the Ref image I_r. However, I_r contains many irrelevant/noisy information to I_{b1} (they are not adjacent/neighboring frames). How to avoid irrelevant referenced information disturbing feature fusion?
2. Performance is promising. But compared to BurstSR methods, RefBSR has Ref frame, and compared to RefSR methods, RefBSR has several adjacent frames. It would be better if compared to Burst + Ref SR methods.

**Suitability:**

3

---

### Official Review · Reviewer_88g2 · 2024-05-24

**Rating:** 3
**Confidence:** 3

**Summary:**

This paper introduces a novel reference-based burst super-resolution (RefBSR) method to combine the strengths of Burst Super-Resolution (BurstSR) and Reference Super-Resolution (RefSR) while overcoming their limitations. RefBSR adaptively fuse the Ref and the burst features to achieve high-quality image reconstruction through the proposed burst feature-aware Ref texture transfer (BRTT) and Ref-burst adaptive feature fusion (RBAF) modules. Additionally, they also provide a new dataset of Ref-burst pairs collected using commercial smartphones for training and evaluation. Experimental results demonstrate that the proposed method outperforms existing RefSR and BurstSR methods.

**Strengths:**

1. The paper is well-written and straightforward to follow.
2. RefBSR effectively integrates the BurstSR and RefSR, leading to superior high-resolution image reconstruction with enhanced detail and texture fidelity.
3. They collect a new dataset consisting of 2,287 pairs of burst frames and Ref images.
4. Extensive experiments demonstrate superior performance.

**Limitations:**

1. As the first paper to use both burst and Ref images, the motivation behind this idea is not sufficiently explained.

2. The experiments are all conducted on the proposed dataset. This raises concerns about the generalizability of this method. It remains to be seen whether the method is still effective on other datasets.

3. The methods for comparison are trained on either burst or Ref images alone. Thus, the fairness of the experiments and the effectiveness of the proposed method need further clarification.

**Suitability:**

3

---

### Official Review · Reviewer_zE9A · 2024-05-25

**Rating:** 4
**Confidence:** 4

**Summary:**

This paper proposes a reference-based burst super-resolution (RefBSR) method that combines the advantages of Burst Super-Resolution (BurstSR) and Reference-based Super-Resolution (RefSR) to enhance the quality of super-resolved images. By harnessing both multiple burst frames and a high-resolution reference image, RefBSR efficiently restores finely textured images that were previously challenging due to factors such as hand tremors. The proposed framework incorporates innovative modules like Ref-burst feature matching and Burst Feature-Aware Reference Texture Transfer (BRTT), alongside Ref-burst Adaptive Feature Fusion (RBAF), which adaptively integrates high-quality features from both reference and burst sources. The proposed dataset consisting of Ref-burst pairs collected from commercial smartphones supports the training and evaluation of this framework, which has demonstrated state-of-the-art performance in comprehensive experiments.

**Strengths:**

1. Combines BurstSR and RefSR methodologies to leverage the strengths of both, enhancing texture restoration in super-resolved images.
2. Provides a dataset specifically designed for RefBSR, allowing for training and comprehensive evaluation of the method.
3. Achieves superior results compared to existing BurstSR and RefSR methods, as demonstrated in extensive experiments.

**Limitations:**

1. The motivation and issues are not clear. The authors state that "if there is no correspondence between the Ref and the low-resolution (LR) images". However, RefSR has considered correspondence matching between them. Could you provide more details on this?

2. In Figure 1, the result of RefSR seems to be problematic. The based frame has vertical textures, but it is removed by RefSR. This may not happen in RefSR.

3. The authors state that the proposed method is the first framework for reference-based burst super-resolution (RefBSR). However, there is one paper [a] also using reference-based techniques. Could you discuss the difference between them?

    [a] Burst Super-Resolution with Diffusion Models for Improving Perceptual Quality

4. In Figure 2, what happens when Reference I_r is replaced with a certain frame in Burst Frames? It is more practical in the real-world setting.

5. In Table 1, the name of the paper [34] sometimes is MRefSR, and sometimes LMR (see Figure 6). The authors should use one name.

6. In Figure 6, there is only one visual result about RefSR. Could you provide more results about RefSR in the main paper?

7. The authors should compare and discuss the efficiency of different methods, including inference time, and model size.

**Suitability:**

2

---

### Meta-Review · Area_Chair_sS7L · 2024-07-04

**Recommendation:** Accept (Poster)
**Confidence:** 5

**Metareview:**

The authors provided a response and all the reviewers are generally positive with respect to the contributions from this paper.

The meta-reviewer agrees with the reviewers and invites the authors to further refine their work for the camera ready by benefiting from the received feedback and integrating contents from the authors' response.